# Prediction of the Remaining Useful Life of Bearings Through CNN-Bi-LSTM-Based Domain Adaptation Model

**DOI:** 10.3390/s24216906

**Published:** 2024-10-28

**Authors:** Feifan Li, Zhuoheng Dai, Lei Jiang, Chanfei Song, Caiming Zhong, Yingna Chen

**Affiliations:** School of Information Engineering, College of Science & Technology, Ningbo University, Ningbo 315000, China; 2211170007@nbu.edu.cn (F.L.); 2311170002@nbu.edu.cn (Z.D.); jianglei2@nbu.edu.cn (L.J.); songhanfei@nbu.edu.cn (C.S.)

**Keywords:** bearing, RUL, CNN, Bi-LSTM, domain adaptation

## Abstract

Predicting the remaining useful life (RUL) of mechanical bearings is crucial in the industry. Estimating the RUL enables the assessment of health bearing, maintenance planning, and significant cost reduction, thereby fostering industrial development. Existing models rely on traditional feature engineering with feature changes because operating conditions pose a major challenge to the generalization of RUL prediction models. This study focuses on neural network-based feature engineering and the downstream prediction of the RUL, eliminating the need for specific prior knowledge and simplifying the development and maintenance of models. Initially, a convolutional neural network (CNN) model is employed for feature engineering. Subsequently, a bidirectional long short-term memory network (Bi-LSTM) model is used to capture the time-series degradation characteristics of the engineered features and predict the RUL through regression. Finally, the study examines the influence of operating conditions in the model and integrates domain adaptation to minimize differences in feature distribution, thereby enhancing the model’s generalizability for the RUL prediction.

## 1. Introduction

Bearing life prediction is a critical technology in the industrial sector, meant to identify bearing health and predict its lifespan early to facilitate planned maintenance and reduce economic losses. The entire life cycle of mechanical bearings is divided into four stages: initial operation, stable operation, progressive wear, failure and end of life [1]. The condition of bearings is intricate and varied, with numerous issues likely to arise throughout their life cycle. Initially, these problems were predominantly addressed manually by experienced technicians. With the advancement of sensors and big data, bearing damage prediction technology has rapidly developed, encompassing condition-based maintenance (CBM) and prognostics and health management (PHM) [2]. Initially applied in the aviation sector, these techniques have gradually expanded to other industrial fields. Existing bearing prediction methods can be broadly categorized into two types: traditional machine learning methods that incorporate prior knowledge [3,4,5,6,7] and deep learning methods primarily driven by data [8,9,10,11,12,13,14,15,16].

Traditional machine learning methods utilize numerical analysis to extract valuable features based on prior knowledge and then select appropriate models for fitting. For example, Loutas et al. proposed an ε-SVR model for monitoring rolling bearing conditions based on wavelet transformation to extract relevant features and Wiener entropy [4]. Qian et al. introduced an RQA-AR model for bearing degradation state extraction using Kalman filtering [5]. Jin et al. proposed an AR model based on extended Kalman filtering [6], while Li et al. presented a degradation-hidden Markov model [7]. Traditional model-based approaches rely on extensive prior knowledge to extract damage feature parameters (for example, kurtosis and time–frequency domain correlation indicators) from data to build corresponding models. However, with the increasing integration and complexity of modern systems, constructing such models has become increasingly challenging.

With the rise in convolutional neural networks (CNNs), recurrent neural networks (RNNs), and long short-term memory (LSTM) networks [17,18,19], researchers have gradually introduced deep networks for deep feature extraction, the joint analysis of previously extracted features, and remaining useful life (RUL) regression prediction [8,9,10,11,12,13], with prior knowledge serving only as an auxiliary for model training. As the data integration capability of deep networks improves, collaborative multi-view prediction has also undergone significant developments. For instance, Ren et al. proposed a deep neural network (DNN) model for multibearing collaborative prediction by introducing a new frequency-domain feature in addition to the traditional time- and frequency-domain features [14]. Although integrating prior knowledge with deep models has enhanced predictive performance, the generalization ability of models based on supervised learning is often limited by factors such as small data volumes and large differences in the distribution of data features under different conditions. Thus, researchers have introduced transfer learning to address these issues. For example, Zeng et al. proposed, in the offline stages, that semi-supervised training be conducted. It is then applied in the online stages for domain adaptation models to predict the RUL [15], with the aim of improving model generalization. However, a lack of data makes model training challenging. To address this issue, He et al. proposed a semi-supervised generative adversarial network (GAN) method for predicting the RUL using fault and suspension histories to generate more data [16].

The correlation method that leverages prior knowledge has demonstrated significant success; however, it still faces certain limitations, including the substantial costs associated with constructing relevant feature projects and the influence of subjective factors on the model [3,4,5,6,7,8,9,10,11,12,13,14,15,16]. So, researchers no longer consider prior knowledge a necessary condition for extracting data features, achieving significant breakthroughs in fields such as image classification and object detection using CNNs [20,21,22,23,24]. Consequently, deeper and broader models have been developed for bearing fault detection. For instance, Wen et al. proposed a CNN model based on LeNet-5 for fault diagnosis [25], which extracts features using a method that converts signals into 2D images, thereby eliminating the need for manual feature engineering. Models, such as the NB-CNN by Chen et al. [26] and DNCNN by Jia et al. [27], have demonstrated excellent performance in damage diagnosis by directly extracting features from raw signals using neural networks. However, in practical engineering applications, the lack of labeled data often complicates fault diagnosis. To address this challenge, researchers have increasingly turned to stacked denoising autoencoders due to their robust feature extraction capabilities and resistance to noise. Wang et al, Saucedo et al. and Chen et al. use autoencoders to mine deep features of the data to build relevant models [28,29,30], these models can effectively learn and represent complex data patterns without requiring extensive prior knowledge, thereby enhancing both the adaptability and performance of the system [28,29,30]. These approaches reduce the reliance on prior knowledge. However, most of these efforts focused on bearing fault identification rather than RUL prediction.

Therefore, this paper proposes a deep learning-based model for predicting bearing life. This model automatically extracts features from bearing data points and reveals the temporal relationships among these points. This paper presents a CNN-Bi-LSTM model framework, where a CNN extracts signal features in discrete time, a Bi-LSTM captures the temporal characteristics of these discrete-time features, and regression predicts the RUL based on the extracted spatiotemporal features. In addition, a domain adaptation module was integrated into the proposed model to enhance its generalization and provide better robustness under different operating conditions. Section 2 introduces the methods used in this experiment, Section 3 presents the research results and discussion, and Section 4 concludes this paper.

## 2. Materials and Methods

### 2.1. Methods

#### 2.1.1. CNN-Bi-LSTM Model

Building on the traditional CNN and LSTM, we propose a neural network model for predicting the RUL of bearings under one operating condition. The proposed framework focuses on modelling and analysing raw data from two perspectives: feature extraction and regression prediction, ultimately forecasting the RUL. While CNN and Bi-LSTM models have been extensively employed across various domains [31,32,33], their integration for RUL prediction presents distinct advantages over conventional methods. Recent studies have successfully implemented the CNN-Bi-LSTM combination in several fields, including raw material inventory management, RUL estimation for lithium-ion batteries, and air quality monitoring [34,35,36]. The demonstrated effectiveness of this fusion model across these domains motivates its application in bearing RUL forecasting. However, unlike these areas, the bearing prediction task involves unique challenges, such as the complexity of mechanical systems operating under multiple conditions, varied specifications, and diverse failure modes. In this context, the present study aims to validate the efficacy of CNN and Bi-LSTM for bearing RUL prediction by leveraging their ability to capture spatiotemporal features. Additionally, domain adaptation techniques are incorporated to address feature misalignment issues arising from multi-condition scenarios. The specific functions of the proposed module are detailed as follows:Feature Extraction and Temporal Dependency: CNN excels at extracting spatial and deep features from raw data, while Bi-LSTM captures sequential dependencies over time. For bearing RUL prediction, this combination allows the model to efficiently extract patterns (via a CNN) and account for their temporal evolution (via a Bi-LSTM). Unlike traditional methods that focus on either spatial or temporal analysis, this approach integrates both.Eliminating the Need for Prior Knowledge: Many existing techniques in bearing fault detection and RUL prediction rely on handcrafted features and domain expertise. The CNN-Bi-LSTM model eliminates this need, automatically learning spatial and temporal features from raw data, reducing bias and saving time.Enhanced Robustness through Domain Adaptation: Incorporating domain adaptation allows the model to handle differences between training (source) and test (target) data distributions, improving generalization. This is critical in industrial applications with varying operating conditions, where traditional models often struggle.

In summary, the novelty lies not only in applying CNN and Bi-LSTM to a new field but also in their synergy to handle both feature extraction and time sequence analysis, combined with domain adaptation to improve model generalization and robustness.

The comparison of the present method with some other recent methods is documented in Table 1. Compared to the seven existing studies, this work distinguishes itself by integrating CNN and Bi-LSTM models for bearing RUL prediction, offering a more comprehensive solution than prior approaches that either focus solely on spatial (CNN) or temporal (LSTM/Bi-LSTM) features. Unlike the works of Guo et al. [34], Mejia et al. [36], and Mao et al. [33], which still rely on some level of handcrafted feature extraction or focus on specific domains like inventory management and rolling bearings, this study eliminates the need for prior domain knowledge, learning features directly from raw data. The incorporation of domain adaptation, which is absent in the works of Rincón-Maya et al. [35], Wang et al. [32], and van Houdt et al. [31], addresses the critical challenge of feature misalignment under varying operating conditions, enhancing the model’s robustness. Thus, the proposed method excels not only in extracting spatiotemporal features but also in improving generalization across diverse and complex industrial scenarios, an area where conventional models, such as the Wiener process used by Zhang et al. [3], often fall short.

The framework of this method is illustrated in Figure 1. The data fragmentation module slices the raw data to expand the dataset. The CNN module extracts features from data points, effectively representing the data after dimension reduction. The Bi-LSTM module captures temporal features between data points and performs regression prediction using the Conv1D module.

#### 2.1.2. Raw Data

The experiment utilized the PHM2012 challenge dataset (FEMTO-ST Bearing Dataset) [37] provided by the FEMTO-ST Institute. This dataset consists of bearing data collected under three operational conditions and sampled at a frequency of 25.6 kHz. Data were recorded every 10 s, resulting in 2560 samples per recording. The specifications of the dataset are listed in Table 2. Appendix E describes the dataset in detail.

The dataset had a sampling interval of 10 s between sample points, with 2560 samples constituting one sample point, representing the features of that point. The sequence of sample collection was used to determine the order of the time-series. The RUL was 339 s for the Bearing1_4 dataset; however, it was rounded to 340 s for prediction convenience. This adjustment introduced some errors but they were manageable. The sample sizes of the datasets are listed in Table 3.

#### 2.1.3. Data Processing

This experiment did not rely on conventional methods for extracting information from the time and frequency domains to evaluate bearing conditions. Instead, the experiment described the vibration state of the bearings in a more intuitive manner by extracting data amplitudes. The results are shown in Figure 2, where labels a, b, c, and d represent the data amplitudes of Bearing1_1, Bearing1_2, Bearing2_2, and Bearing3_2, respectively. The complete amplitude chart is provided in Appendix A. As illustrated in Figure A1 (Appendix A), there are significant variations in vibration data across different operational conditions; even within identical conditions, notable discrepancies can be observed in the vibration data.

The amplitude plots of the data clearly reflect the vibration intensity, providing additional insights into the experimental results.

Using the Pandas library, we have selected data in both the horizontal and vertical directions. The dataset is already clean, with no errors or duplicate entries, and thus requires no additional preprocessing. To eliminate the effects of varying scales, accelerate computations, enhance model accuracy, and mitigate the impact of outlier samples, we applied min–max normalization to the data. Furthermore, the data were labeled based on the time of collection, and the labels were normalized to a range of 0 to 1 using the min–max normalization method.

Due to the limited volume of the dataset, traditional data augmentation techniques may introduce unnecessary noise into the generated data. Therefore, we collected one sample point (each containing 256 vibration features) within 0.1 s at every 10-s interval, enabling us to segment the data accordingly. Within this extremely short time span, the vibration state of the bearing remains stable, allowing each of the resulting 10 segments to represent the vibration state at a specific point. This approach effectively expanded the dataset tenfold, alleviating the issue of insufficient data for comprehensive model training. Finally, we used the average of these 10 segments as the final representation of the data.

#### 2.1.4. Localized Feature Extraction

A CNN was used to extract the localized features of the bearing vibration signals. A diagram of the CNN architecture is depicted in Figure 3.

In the CNN architecture, we employed three convolutional layers, each followed by an activation function, batch normalization, and max pooling to process the data. Vibration data were primarily collected from sensors in horizontal and vertical directions. In the first layer, we used (1, 8) convolutional kernels to capture more global information and structural features from the data. Additionally, to improve the generalization of the model, we increased the number of channels from 1 to 16, allowing for a more comprehensive exploration of data point information. Given the characteristics of the vibration data, we utilized a rectified linear unit (ReLU) as the activation function to transform the post-convolution data into a new representation. This approach assisted the model in extracting and representing various feature types [8] to prevent overfitting, gradient vanishing, or exploding. Additionally, to accelerate training while improving model generalization, we applied batch normalization to further process the data. Max pooling was used to reduce dimensionality and extract higher-level feature representations from the data, thereby enhancing computational efficiency, reducing model complexity, and increasing model translational invariance. A similar process was applied to the second and third layers. However, for these convolutional layers, we utilized 1 × 4 kernels to effectively explore local information and capture detailed features within the data.

After processing through the CNN layers, the model captures high-level feature representations of the time series, while reducing the dimensionality of the data. This reduction helps decrease the computational complexity of the bidirectional LSTM (Bi-LSTM) and improves the overall computational efficiency of the model.

#### 2.1.5. Long Timing-Related Feature Extraction

To explore the temporal information between the data points, we designed a two-layer Bi-LSTM architecture. The utilization of a two-layer Bi-LSTM enables hierarchical feature learning, addresses long-term dependencies between data points, and reduces the likelihood of gradient vanishing leading to improved performance. While the model effectively captures temporal information between data points, it can increase computational complexity and lead to overfitting. Therefore, within the two-layer Bi-LSTM model, we incorporated a dropout function to mitigate the risk of overfitting. The extracted features were then used for regression prediction. The architecture of the Bi-LSTM is depicted in Figure 4. The input feature dimension is set to 160, and the hidden unit size is determined to be 64 through grid random search optimization. To mitigate overfitting, the dropout rate is set at 0.5.

#### 2.1.6. Regression Prediction

The features extracted by the Bi-LSTM did not exhibit strong linear relationships. Instead of using traditional fully connected layers for RUL prediction, we propose the use of one-dimensional (1D) convolutional prediction. One-dimensional convolutional layers possess strong nonlinear extraction capabilities and are widely used in processing 1D sequence data, such as text classification, speech recognition, signal processing, and time-series analysis [38]. The use of 1D convolution allows for the extraction of more local information among the features. Moreover, to reduce the model complexity, alleviate the computational burden, and mitigate the risk of overfitting, we utilized only one layer of 1D convolution for the regression prediction. Consequently, its configuration is defined with an input channel of 1, an output channel of 1, a convolution kernel size of (1, 128), and a stride of 128.

#### 2.1.7. Domain Adaptation Model Framework

Domain adaptation can solve the generalization problem of the model on different data distributions. We integrated the domain adaptation function into the trained CNN-Bi-LSTM model, mainly by connecting the CNN-Bi-LSTM module to the codomain adaptation module, and copied the features extracted by CNN-Bi-LSTM into two copies, one into the regression prediction module and the other into the domain adaptation module. The CNN-Bi-LSTM model itself needs to complete the task of RUL prediction, and after integrating the domain adaptation function, it also needs to complete the task of domain classification (distinguishing the source-domain from the target-domain). The model adopts the joint training method, and the loss consists of the prediction loss of the regression prediction layer and the classification loss of the domain adaptation layer (the total loss is formally expressed in Equation (Equation 1), where *N* denotes the number of samples, ${\hat{y}}_{i}$ represents the predicted values, and $y_{i}$ corresponds to the ground truth labels). Additionally, *M* refers to the number of samples used for domain classification, with ${\hat{d}}_{i}$ being the output of the domain classifier and $d_{i}$ representing the true domain labels. We adopt the adversarial domain adaptation method to train the domain adaptive module, through which the model will be as close as possible to the source-domain and the target-domain in the process of training (by using crossentropy loss function and gradient inversion method, the model cannot distinguish the source-domain from the destination domain) to achieve feature alignment. After domain adaptation, the difference of the input features into the regression prediction module is small, so the model can obtain better results when predicting the data with different feature distributions [39].

Based on the experimental results described above, we observed significant differences in the vibration data from the bearings under various operating conditions, including the same operating conditions. To enhance the robustness of the previously proposed model and enable it to adapt to various vibration data scenarios, we incorporated domain adaptation through transfer learning, thereby making the entire model framework more resilient.

The model architecture used to incorporate domain adaptation into the base model is illustrated in Figure 5.

The domain adaptation module consists of three linear layers. Before the features entered the domain adaptation module for training, gradient reversal processing was applied. This approach helps the model to adapt the features from the source-domain to match those of the target-domain, while minimizing feature discrepancy [39]. The functionalities of the other modules in the model remained consistent with those of the base model. The target-domain data used in the domain adaptation module were not included in the RUL training process. Finally, we utilized the Jensen–Shannon (JS) divergence Equation (2) to assess the effectiveness of the domain adaptation module. A larger JS divergence indicates greater differences in the feature distribution between datasets, whereas a smaller JS divergence indicates smaller differences.
(1)$$L_{\mathrm{total}}=\frac{1}{N}\sum_{i=1}^{N}{\left({\hat{y}}_{i}-y_{i}\right)}^{2}-\frac{1}{M}\sum_{i=1}^{M}\left[d_{i}\log\left({\hat{d}}_{i}\right)+\left(1-d_{i}\right)\log\left(1-{\hat{d}}_{i}\right)\right]$$
(2)$$JSD\left(P\|Q\right)=\int p\left(x\right)\log\frac{p(x)}{\frac{p(x)+q(x)}{2}}dx+\int q\left(x\right)\log\frac{q(x)}{\frac{p(x)+q(x)}{2}}dx$$

### 2.2. Materials

#### 2.2.1. Motivation

Most of the traditional RUL prediction methods preprocess the data through feature engineering, extract the time–frequency domain features of the data according to relevant prior knowledge, and then predict the bearing RUL after model training [8,9,10,11,12,13,14,15,16]. Different methods extract corresponding features according to different prior knowledge, which undoubtedly increases the cost of feature engineering. At the same time, because the time–frequency domain features extracted by different feature projects are different, the relevant features extracted may not be applicable to different models. Therefore, this paper abandons the method of extracting relevant time–frequency domain features based on prior knowledge, and directly extracts the deep features of data through the neural network model. This method can reduce the cost of feature engineering without selecting time–frequency domain features according to relevant prior knowledge, so that the model can be better applied to different datasets and then applied in industry [34]. The CNN model can extract the deep features of the image, and the image itself belongs to a data type. Therefore, this paper uses a CNN model to mine the deep features of a single data point. At the same time, bearing data belong to time sequence data, and time sequence information is hidden between points, and the Bi-LSTM model has certain advantages in extracting time sequence information [40]. Therefore, this paper adopts the Bi-LSTM model to mine the time sequence information between data points. Different situations may occur in the running process of bearings, which may lead to large differences in data collected under different working conditions or even under the same working condition. This paper uses the domain adaptation method in transfer learning to solve this problem. Therefore, a domain adaptive model based on CNN-Bi-LSTM is proposed to predict bearing RUL.

#### 2.2.2. Reproducibility

The algorithm of the proposed method is shown in Algorithm 1. Algorithm 1 provides a detailed description of the proposed method: CNN and Bi-LSTM are effectively integrated (with the batch input to CNN serving as the sequence length for Bi-LSTM) to address the high cost and subjectivity of traditional feature engineering. Additionally, the integrated domain adaptation module resolves the challenge of significant feature distribution differences across various working conditions. Unlike other algorithms [8,9,10,11,12,13,14,15,16], this approach eliminates the need for feature extraction based on prior knowledge, reducing both feature engineering costs and the impact of subjective bias, while also addressing feature distribution discrepancies under diverse conditions. The advantages and limitations of the proposed method will be further discussed in the Conclusions section.
**Algorithm 1** The algorithm of the method proposed in this paper  1:**Input**: Raw data  2:**Output**: RUL  3:Data processing(Process the raw data and label it accordingly)  4:Initialize the model parameters  5:**for** each train **do**  6:   CNN mines the deep features of the data points  7:   Bi-LSTM mines timing information between data  8:   The regression layer makes the prediction(Get the RUL of the data point)  9:**end for**10:Save model11:Freeze the CNN and Bi-LSTM modules of the saved model, retrain the regression layer module and add the domain adaptation module12:**Input**: Raw data13:**Output**: RUL14:Initialize the model parameters15:**for** each train **do**16:   CNN mines the deep features of the data points17:   Bi-LSTM mines timing information between data18:   The regression layer makes the prediction(Get the RUL of the data point) and The domain adaptation layer classifies data domains (two categories: source domain and target domain)19:**end for**20:Calculate the RMSE for each dataset

#### 2.2.3. Description of Models Used

In the field of bearing RUL prediction, mean absolute error (MAE) and root mean square error (RMSE) are usually used as evaluation indexes, and the lower the value, the better the performance of the model. The formula of MAE is shown in Equation (Equation 3), and the formula of RMSE is shown in Equation (Equation 4) [41]:(3)$$\mathrm{MAE}=\frac{1}{n}\sum_{i=1}^{n}\left|y_{i}-{\hat{y}}_{i}\right|$$
(4)$$\mathrm{RMSE}=\sqrt{\frac{1}{n}\sum_{i=1}^{n}{\left(y_{i}-{\hat{y}}_{i}\right)}^{2}}$$
where *n* represents the number of samples, $y_{i}$ represents the actual RUL of the data point, and ${\hat{y}}_{i}$ represents the RUL of the data point predicted by the model.

#### 2.2.4. Evaluation Methods

MAE evaluates the average magnitude of the errors between the predicted and actual values, offering a direct measure of prediction accuracy. It is particularly useful for applications where each error is equally important, as it does not square the differences, ensuring that all deviations contribute proportionally to the overall error [41].

RMSE, on the other hand, penalizes larger errors more heavily due to the squaring of differences between predicted and actual values. This makes it sensitive to outliers, meaning it highlights models that produce large deviations from actual RUL values. Therefore, RMSE is often preferred when large errors are particularly undesirable, and reducing them is a priority [41].

Both metrics provide complementary perspectives: MAE gives a straightforward view of the overall error, while RMSE emphasizes the significance of larger mistakes in the prediction [41].

In evaluating models for predicting RUL, traditional performance metrics like RMSE, MAE, MSE (Mean Squared Error), and MAPE (Mean Absolute Percentage Error) are commonly used. MAPE is often avoided due to several key limitations. First, MAPE becomes problematic when actual values approach zero, leading to disproportionately large or undefined percentage errors. Second, MAPE tends to over-penalize small actual values, inflating errors even when the absolute difference is minimal. Additionally, it is insensitive to large errors for larger actual values, which may conceal significant prediction inaccuracies. MAPE also fails to symmetrically handle over- and under-predictions and cannot accommodate negative values, limiting its applicability in many contexts such as temperature or profit forecasting. Therefore, alternatives like MAE or RMSE are preferred in many applications [41]. At the same time, there is essentially no difference between MSE and RMSE, so only RMSE can be used. In this paper, the results of MAE and RMSE obtained were compared with those of others, and the results of different models were compared through ablation experiments to determine the validity of the proposed method.

## 3. Results and Discussion

### 3.1. Experimental Results of CNN Data Point Feature Extraction

In this paper, according to the multi-layer feature extraction model proposed by [8], a three-layer CNN model was built to initially extract the deep features of data points to represent a single data point. The model was validated using datasets from the Bearing Data Center at Case Western Reserve University (CWRU) [42]. Table 4 shows the relevant parameters selected using the grid search method and the relevant hyperparameters finally determined by the model through the grid search method. The dataset is shown in Table 5. All CWRU data are shown in Table 6. It can be seen from the table that there are four types of data under the four working conditions: normal, inner ring damage, outer ring damage, and ball damage. We represent all damage data as 1 and normal data as 0. At the same time, the data collected on the corresponding bearing in each file is for a period of time, so we take 256 data as one data point to divide all the data into multiple sample points for model training.

As can be seen from Table 5, the accuracy of the model on the training set and the test set reaches 100%. To further test the performance of the model, we tested the datasets under conditions 2, 3, and 4 with the model, and recorded the experimental results in Table 6. As can be seen from the table, for datasets under different conditions, the prediction accuracy of the model for fault data reached 100%, and the accuracy was also high for normal datasets (due to the small amount of data of normal samples, the prediction accuracy of the model for normal samples under different conditions did not reach 100%). At the same time, it can be seen from Figure 6 that when the model is trained for about 140 rounds, the model becomes stable and the prediction accuracy of the training set and test set reaches 100%. Combined with Table 5 and Figure 6, it can be seen that the model can mine the deep features of a single data point.

To further verify the effectiveness of the model in extracting the deep features of data points, we used the FEMTO-ST bearing dataset to train the above model. The FEMTO-ST bearing dataset is a life-cycle dataset, and the data themselves do not have fault labels, so we label the data according to the bearing fault point (FOT) proposed in [9]. At the same time, the amplitude of the data can better reflect the running state of the bearing, so we select the relevant FOT points according to the amplitude of the data. The amplitude of the data is shown in Figure 7. Data before the FOT point are part of a normal sample (assigned 0), while data after the FOT point are part of a fault sample (assigned 1). The FOT points constructed by the two methods are shown in Table 7, and the relevant experimental results are recorded in Table 8.

As can be seen from Table 8, the prediction performance of the model is significantly better according to the labels on the FOT points determined by the amplitude. In order to further determine the influence of FOT points, we plotted the predicted values of Bearing1_5, Bearing1_6, and Bearing1_7 models (three datasets with low accuracy) and the FOT points determined by the two methods in Figure 8. The full amplitude FOT point plot is documented in Appendix B. As can be seen from Figure 8, the FOT points determined based on the amplitude are closer to the predicted results of the model. From Figure A2, it is evident that when the FOT points for the data (Bearing1_1–1_4) determined by both methods exhibit minor differences, the model’s predictive results align closely. However, when substantial discrepancies arise for FOT points concerning the data (Bearing1_5–1_7), those identified through the amplitude method are more representative of actual FOT points.

From the experimental results, the FOT points determined by the amplitude can better reflect the performance of the model. The original data used in this experiment are unprocessed, while the data used by [9] are the time–frequency domain data obtained after preprocessing. The different feature distributions of the data lead to certain differences in the results. Through the above experiments, it can be determined that the built model can mine the deep features of a single data point, so that it can better represent the data point.

### 3.2. CNN-Bi-LSTM Model Experimental Results

In this experiment, trials were conducted using a model framework without domain adaptation. The method of using random search parameters was employed, ultimately determining the optimal parameters, which are listed in Table 9. As can be seen from Table 9, we used the Adam optimizer to adjust the learning rate, adjust the momentum, normalize the parameters, and prevent the model from overfitting. According to the characteristics of the label, we used the mean square error (MSE) as the loss function. To ensure a seamless connection between the CNN and LSTM layers, we used the batch size from the CNN as the sequence length for the LSTM network layer. To maintain the continuity of the time-series, the batch size was sequentially rather than randomly partitioned. The entire experiment was conducted on a desktop computer equipped with a 12th Gen Intel(R) Core(TM) i5-12500 processor, running at 3.00 GHz, and utilizing the Windows 11 operating system. The model comprised a total of 217,519 parameters, and the training duration was approximately 70 s.

The experimental results are shown in Figure 9, where a–h represent the RUL for Bearing1_1, Bearing1_2, Bearing1_3, Bearing1_4, Bearing2_1, Bearing2_2, Bearing3_1, and Bearing3_2, respectively. Model predictions for all datasets in the three conditions are recorded in Appendix C. Figure A1 (Appendix A) highlights distinct differences in vibration data between datasets under operational conditions one and two, while patterns observed under operational condition three bear a greater resemblance to those seen in condition one. Consequently, as depicted in Figure A3 (Appendix C), optimal predictive performance is achieved in the dataset from condition one; near-optimal performance is noted for condition three’s dataset, whereas a marked decline in predictive accuracy occurs with respect to condition two’s dataset. Furthermore, Figure A1 indicates that vibration data from Bearing1_3, Bearing1_4, and Bearing1_7 closely correspond with those from Bearing1_1 and Bearing1_2 under condition one. This correlation is reflected by a high predictive accuracy for these bearings as shown in Figure A3. In contrast, vibration data from Bearing1_5 and Bearing1_6 diverge from this trend, resulting in only sub-optimal predictions for these bearings.

In this experiment, the training set consisted of datasets Bearing1_1 and Bearing1_2 under operating condition one (OC1), while the remaining datasets served as the test set. Due to the varying lifespans of the datasets, direct lifespan values could not be used as training labels. Instead, we utilized a Health Indicator (HI) normalized between 0 and 1 for model training. From Figure 9, it can be observed that the model performed well, with the training dataset closely fitting the model. However, there were significant errors toward the end of the data due to limitations in the network structure. To ensure proper training, the dataset could not be fully partitioned, resulting in the exclusion of end data points from the training. Consequently, the prediction performance for these end data points was sub-optimal, although the errors were controlled within an acceptable range. Furthermore, the performance of the test set for OC1 was notably better than those for OC2 and OC3. As shown in Figure A1, there were significant variations in the vibration data between different operating conditions and within similar conditions. The test set for OC1 exhibited satisfactory results, whereas for OC2, the model predicted stable labels during certain periods, indicating consistent degradation trends and minimal vibration fluctuations. Similarly, this pattern was observed for OC3. Overall, the test set predictions under the three operating conditions were satisfactory. The performance of the end data points was sub-optimal, similar to that of the training set, although within an acceptable range.

Table 10 presents the evaluation metrics used to compare our model with a published I-DCNN [43] model and MCNN [44] model (note that the MAE and RMSE for Bearing1_3–1_7 were not provided in the original paper of MCNN). It can be observed from the table that our method achieved better results for most of the tested samples.

Time-series features play an important role in RUL prediction. Therefore, we use different temporal basic models (such as BI-LSTM, LSTM, RNN, Bi-RNN, GRU, BI-GRU, etc.) to mine temporal features between data for RUL prediction. We only replace the network layer in the model that extracts the temporal features between data, and keep other parameters unchanged to carry out relevant experiments. The experimental results are recorded in Table 11. At the same time, all datasets under the three working conditions obtained by the model are recorded in Appendix D.

As can be seen from Table 11, the dataset with the OC1 has the best prediction effect on the Bi-LSTM model, and its NRMSE is 0.1556 and NMAE is 0.1201. To effectively convey the predictive performance of various models in estimating the RUL of bearings, we visualized the results presented in Table A1 (Appendix D) using box plots. Each value in Table A1 represents either the MAE or RMSE for different models across multiple datasets. Figure 10 (MAE) and Figure 11 (RMSE) provide a comparative performance analysis of these models. From the box plots, it is evident that the CNN-Bi-LSTM model exhibits the lowest mean values for both MAE and RMSE across all datasets. Although the mean value of CNN-Bi-GRU is comparable to that of CNN-Bi-LSTM, its larger variance suggests greater instability within this model. The MAE and RMSE box plots further illustrate that the CNN-Bi-LSTM consistently outperforms other models regarding predictive accuracy across most datasets. Overall, the CNN-Bi-LSTM model distinguishes itself with superior performance on both MAE and RMSE metrics, confirming its robustness across diverse datasets. So, we finally use the Bi-LSTM neural network model to mine the timing features between the data.

In addition, we observed a significant increase in the test set metrics for OC2 and OC3, indicating a decrease in predictive performance. This highlights the impact of operating conditions on the generalizability of the model. Beyond the training datasets (Bearing1_1 and Bearing1_2), we selected two samples from the test sets for each OC (Bearing1_3, Bearing1_4, Bearing2_1, Bearing2_2, Bearing3_1, and Bearing3_2) and examined the feature distribution using t-distributed stochastic neighbour embedding (t-SNE) downscaling. The results are displayed in Figure 12.

We can observe from the feature distribution plots that the feature distribution of the test set closely aligns with that of the training set. However, certain datasets show a high concentration of points, suggesting that the bearings were in a stable state during periods with similar vibration data. Consequently, the predictions during these periods yielded similar results, reflecting a stable operational state of bearings. Meanwhile, it can be seen from the figure that some feature regions of Bearing1_1 and Bearing1_2 are similar, but their RUL values are completely different. As a result, the model cannot accurately predict the RUL value on the test set, which reduces the performance of the model.To further quantify the results, we computed the JS divergence to compare the distribution similarity of the different datasets. The results are summarized in Table 12.

As can be seen from Table 12, the JS divergence between Bearing1_3 and Bearing1_2 is small and the similarity of their feature region is high; also, their prediction performance is good. However, the JS divergence between Bearing2_1 and the two training sets is neither large nor small, and the prediction result is sub-optimal. At the same time, the similarity of feature regions between Bearing1_1 and Bearing1_2 is high. Figure 10 further supports the effectiveness of the model in predicting the features of the dataset. In addition, Table 12 confirms significant differences between the source- and target-domains, which lead to sub-optimal experimental results.

To enhance the performance of the base model, reduce the disparities between the source- and target-domain feature distributions, and achieve more accurate predictions, we employed domain adaptation methods to refine the model and improve its predictive effectiveness.

### 3.3. Domain Adaptation Model’s Experimental Results

In this experiment, we implemented an adversarial domain adaptation method based on the aforementioned base model, as depicted in Figure 5. From the experimental results mentioned earlier, we found that the regression layer was the main factor affecting the prediction result. Therefore, we kept the parameters of the CNN and LSTM layers unchanged while resetting the parameters of the regression prediction network layers for retraining. The basic experimental information is summarized in Table 13.

The evaluation metrics following the application of domain adaptation are presented in Table 14. As shown in the table, the MAE and RMSE for Bearing1_1, Bearing2_1, Bearing3_1, and Bearing3_2 significantly decreased, indicating improved predictive accuracy (as lower RMSE values correspond to higher accuracy). In particular, these bearings exhibited a substantial reduction in error, suggesting that the domain adaptation process effectively aligned the feature distributions and enhanced model performance across various operating conditions. However, for Bearing2_2, the changes in MAE and RMSE were minimal, implying that domain adaptation did not have a noticeable impact on this specific bearing. Despite this, the overall results indicate that the domain adaptation approach was successful in enhancing the model’s predictive capability for most bearings, particularly those with initial misalignment in feature distributions.

Therefore, it can be demonstrated that domain adaptation techniques can minimize the disparities in feature distributions and improve model performance. Additionally, when a bearing is in a stable state, the collected vibration data are similar, leading to similar RUL values predicted by the model.

Table 15 presents the evaluation metrics used to compare the performance of our model with the CNN-Bi-LSTM model, the published I-DCNN model [43], and the MCNN model [44]. The table indicates that, compared to the CNN-Bi-LSTM model, the MAE and RMSE for Bearing1_5 in the proposed model show no significant change, while the MAE and RMSE for the other datasets show a decrease. Additionally, when compared to the I-DCNN and MCNN models (note that the MAE and RMSE for Bearing1_3–1_7 were not provided in the original paper of MCNN), the proposed model exhibits a significantly lower NRMSE. These results demonstrate that the CNN-Bi-LSTM model, when integrated with domain adaptation, effectively addresses the issue of feature misalignment between the source and target-domains, thereby enhancing generalization across different operating conditions. This improvement highlights the critical role of domain adaptation in enhancing prediction accuracy, particularly in multi-condition scenarios.

## 4. Conclusions

This paper proposed a neural network-based method for predicting the RUL. This method leveraged a CNN to extract relevant features from data points and utilized a Bi-LSTM network to capture temporal relationships among the data, thereby replacing traditional feature engineering based on prior knowledge. Domain adaptation techniques were employed to address significant data differences across multiple operating conditions. As illustrated in Table 10, the CNN-Bi-LSTM model has demonstrated superior performance compared to both I-DCNN and MCNN, achieving an NRMSE of 0.1894, which significantly outperforms I-DCNN’s 0.2890 and DCNN’s 0.3805. Furthermore, when enhanced with a domain adaptation module, the CNN-Bi-LSTM model attains an NRMSE of 0.1870, as recorded in Table 15. This performance is markedly better than that of I-DCNN and DCNN, indicating a notable improvement over the standard CNN-Bi-LSTM model. Experimental validation confirmed the effectiveness, correctness, and practicality of the proposed approach. The results demonstrated that this method can accurately predict the RUL and is suitable for industrial applications.

Despite the promising results, the proposed method has several limitations. First, bearing degradation typically occurs after a certain period of operation, a factor not accounted for in this study. This omission creates inconsistencies between data points and their labels, potentially reducing the model’s predictive accuracy. Although data slicing was used to expand the dataset, it remains relatively small, limiting the model’s ability to fully capture and explore complex data features.

In the joint training of the CNN and Bi-LSTM, the batch input of the CNN is used as the sequence input for the Bi-LSTM. While the model trains successfully, the CNN input is partitioned sequentially without shuffling, which may limit the effectiveness of feature extraction. Additionally, the Bi-LSTM input batch size cannot always be reduced to 1, which may restrict the model’s ability to capture temporal information.

Moreover, due to the network’s structural design, tail-end data are ignored during training, resulting in reduced model performance when predicting these data. In this study, the total loss is a combination of the domain adaptation module and 1:1 regression predictions. While this improves performance, further enhancement may be achieved by defining weighted loss functions with varying proportions or by exploring new hyperparameters to better optimize model training.

To address the limitations of the current models, future research will focus on identifying the initial point of azimuth degradation (FOT) to more accurately capture the onset of data degradation. To overcome data scarcity, generative networks will be employed to synthesize large datasets within acceptable noise thresholds, or data augmentation techniques will be applied to enrich the data. Additionally, future efforts will aim to optimize network architectures for a more effective utilization of tail-end data or adopt distributed training strategies to enhance computational efficiency. Further exploration of a more comprehensive domain-adaptive loss function will also be conducted to improve overall model performance.

## Figures and Tables

**Figure 1 sensors-24-06906-f001:**
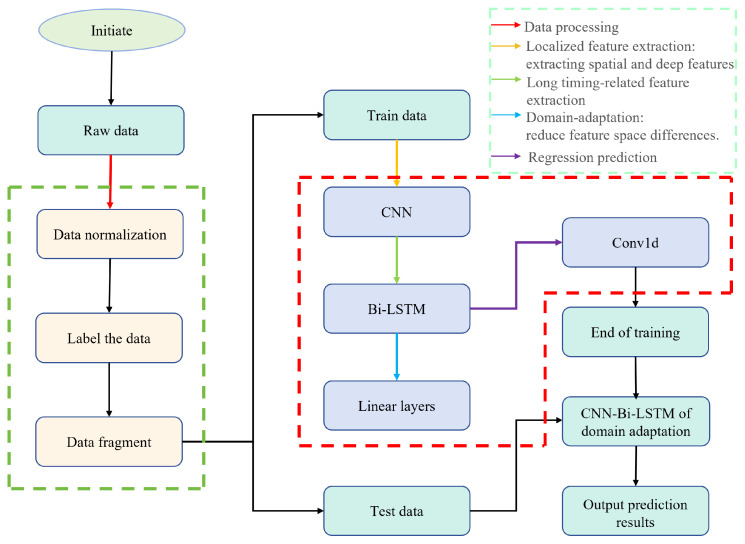
The proposed experimental method is illustrated through a block diagram, showcasing the sequential steps and components involved in the process. The legend in the upper right explains the meaning of the different colored arrows.

**Figure 2 sensors-24-06906-f002:**
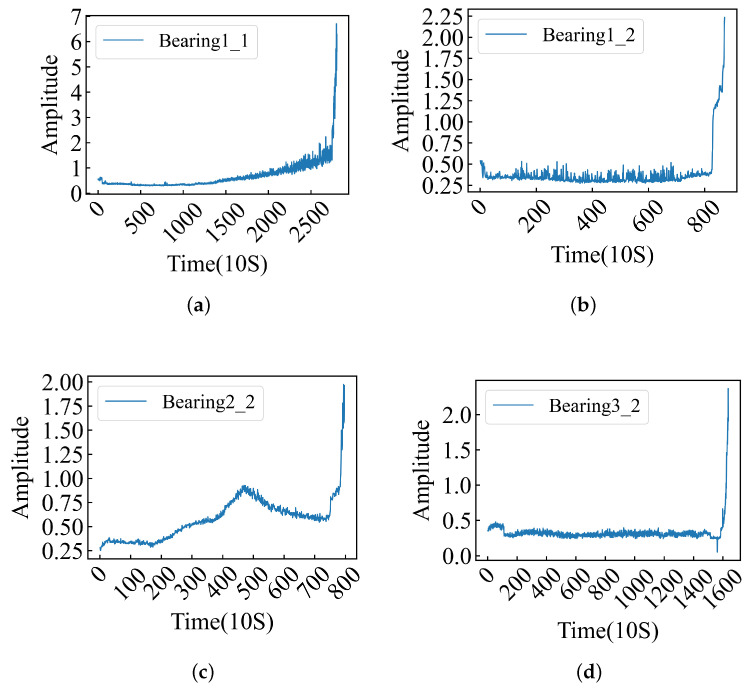
The amplitude bearing data provide an in-depth understanding of the vibration magnitude and intensity experienced by bearings, serving as auxiliary conditions for RUL. (**a**–**d**) Bearing1_1, 1_2, 2_2 and 3_2 amplitudes.

**Figure 3 sensors-24-06906-f003:**
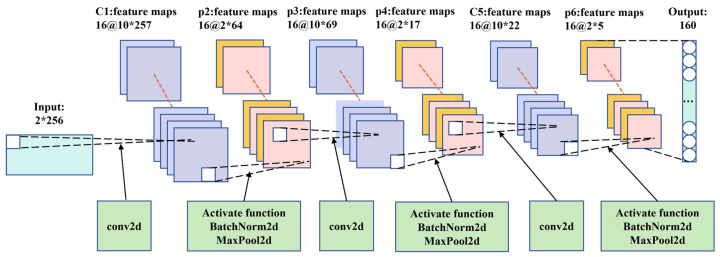
The CNN structure diagram intuitively illustrates the changes in data dimensions within the network architecture.

**Figure 4 sensors-24-06906-f004:**
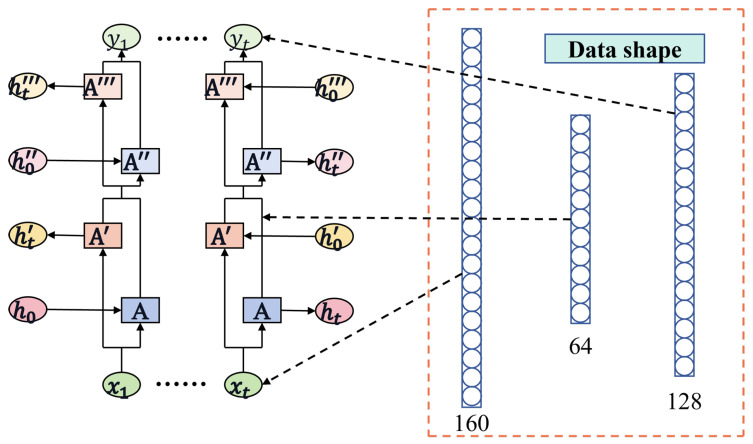
The Bi-LSTM structure diagram provides a visual representation of the architecture and flow of information in a Bi-LSTM neural network.

**Figure 5 sensors-24-06906-f005:**
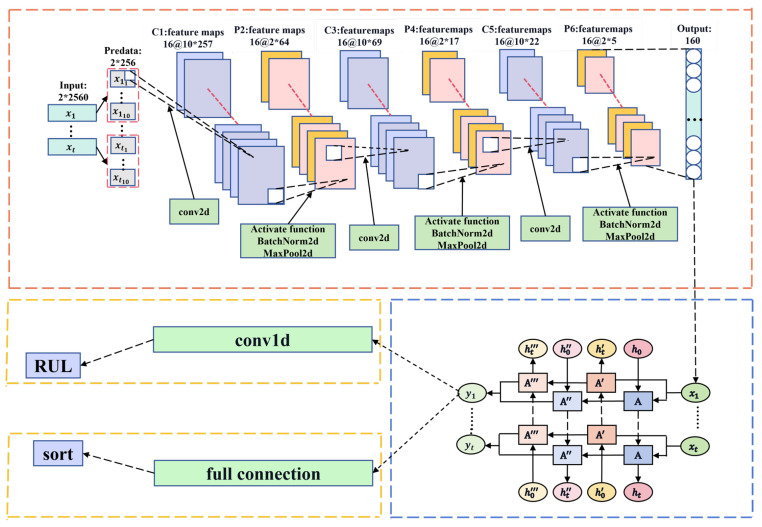
The domain adaptation structure diagram clearly illustrates how to integrate domain adaptation modules on top of the existing network architecture.

**Figure 6 sensors-24-06906-f006:**
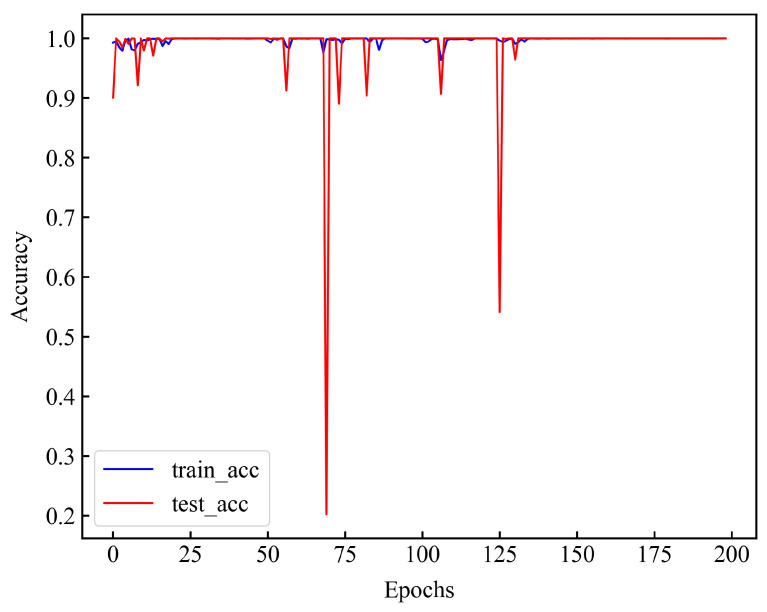
Correlation accuracy of the CNN model, where blue represents the accuracy of the training set and red represents the accuracy of the test set.

**Figure 7 sensors-24-06906-f007:**
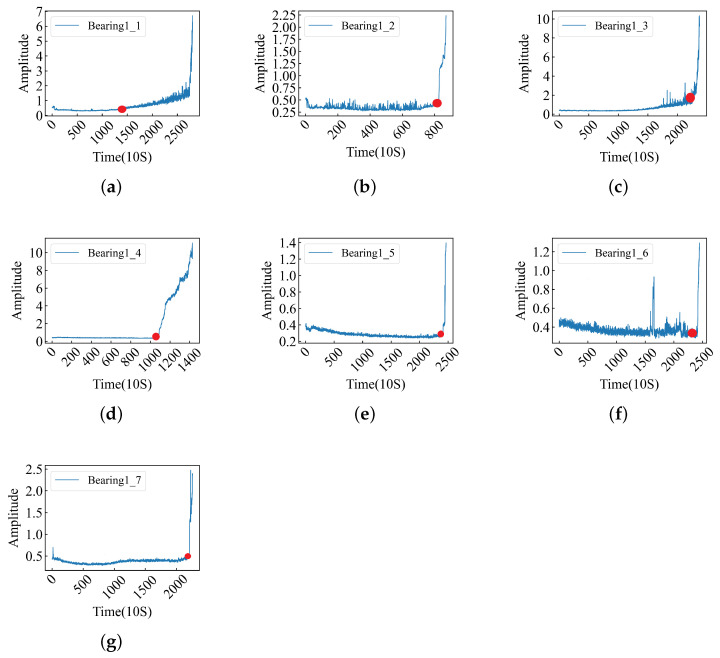
The amplitude of the raw data, where the red circle represents the FOT points. (**a**–**g**) Bearing1_1–1_7 amplitudes and FOTs.

**Figure 8 sensors-24-06906-f008:**
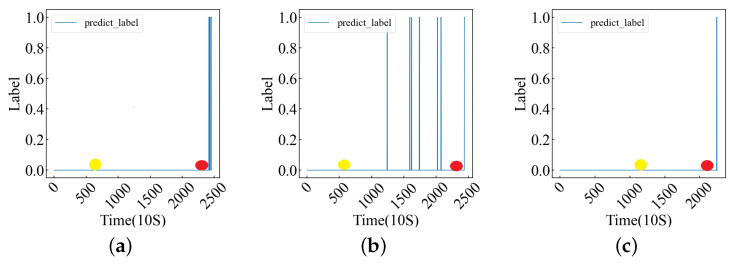
The model’s predicted values, where the red dots represent FOT points based on amplitude and the yellow dots represent FOT points based on [9]. (**a**–**c**) Bearing1_5, 1_6, 1_7 predict_labels and FOTs.

**Figure 9 sensors-24-06906-f009:**
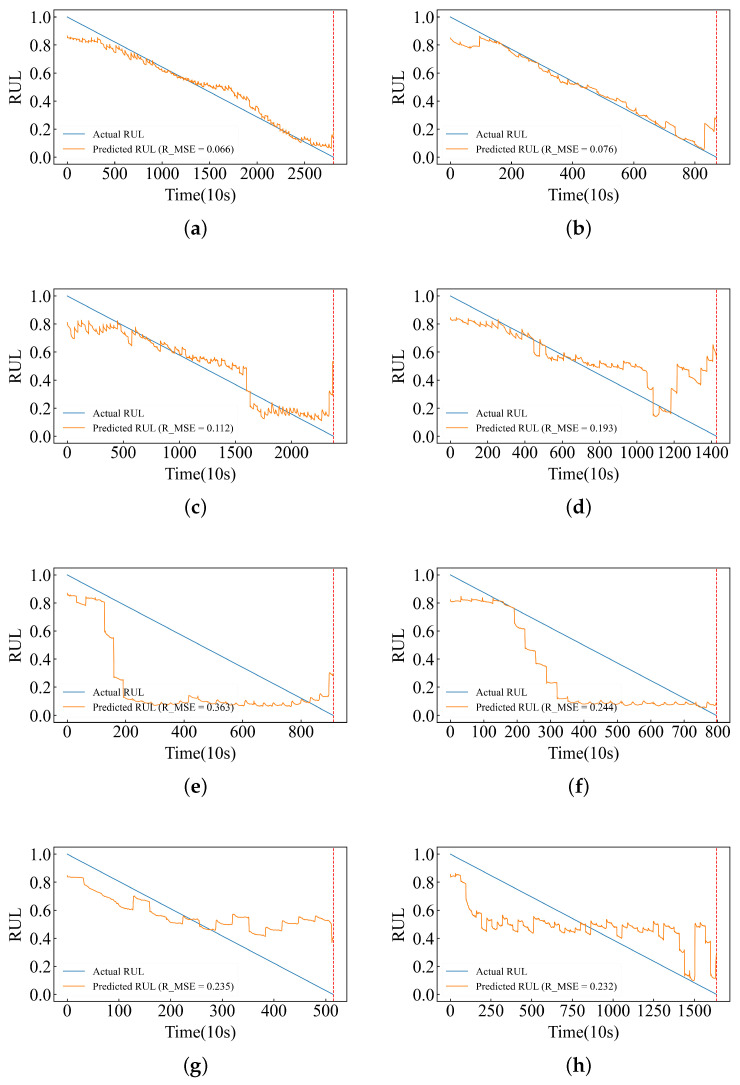
For RUL prediction under various conditions, Bearing1_1 and 1_2 under OC1 are used as training sets. The test set is as follows: Bearing1_3 and 1_4 under OC1, Bearing2_1 and 2_2 under OC2, and Bearing3_1 and 3_2 under OC3 (corresponding to (**a**–**h**) in the figure). The yellow line indicates the predicted RUL, the blue line indicates the actual RUL, and the red line indicates the end of life. (**a**–**h**) Bearing1_1, 1_2, 1_3, 1_4, 2_1, 2_2, 3_1, 3_2 RUL.

**Figure 10 sensors-24-06906-f010:**
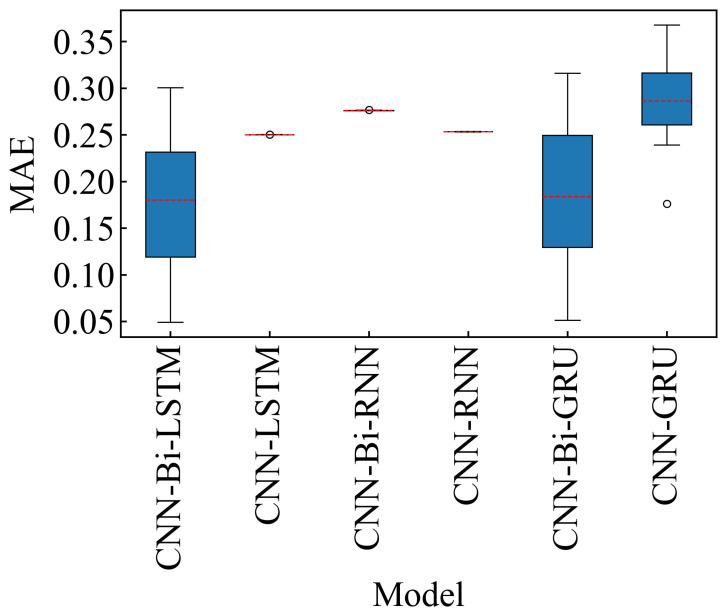
MAE of different models on Bearing1_1–3_3. The red dashed line represents the mean of the MAE of the datasets. Black circles indicate mild outliers.

**Figure 11 sensors-24-06906-f011:**
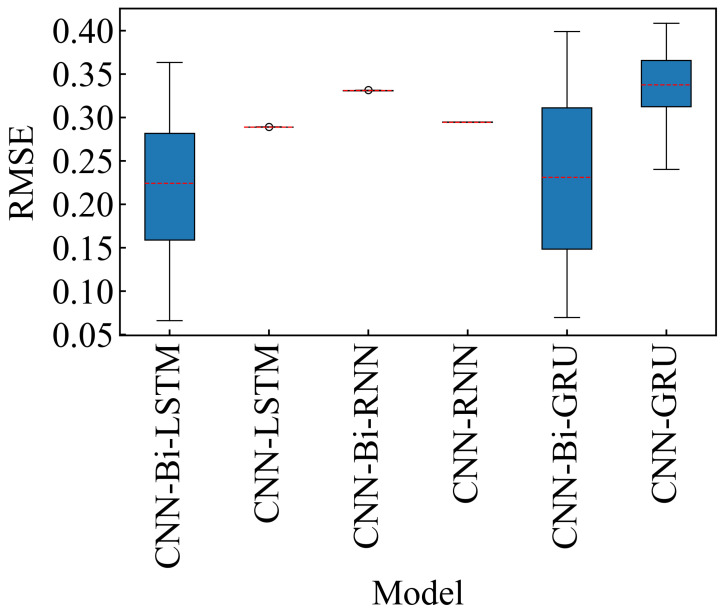
RMSE of different models on Bearing1_1–3_3. The red dashed line represents the mean of the RMSE of the datasets.Black circles indicate mild outliers.

**Figure 12 sensors-24-06906-f012:**
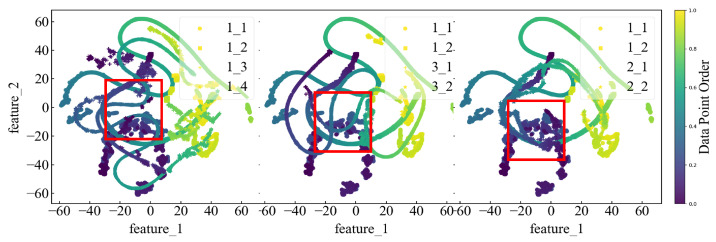
Feature distribution of samples in different OCs. Each test group (Bearing1_3, 1_4, 2_1, 2_2, 3_1, and 3_2) was compared with training groups (Bearing1_1 and 1_2). Samples in the same OC are compared in one subfigure. Red box lines indicate similar feature areas.

**Table 1 sensors-24-06906-t001:** Comparison of the proposed method with recent methods.

**Reference**	**Application Area**	**Main Problem**
Guo et al. [34]	Warehouse management for silica powder	Predicting the amount of silica powder moving in and out of a warehouse
Rincón-Maya et al. [35]	Lithium-ion battery life prediction	Remaining useful life (RUL) prediction of lithium-ion batteries
Mejia et al. [36]	Time-series prediction	Time-series prediction using neural network-based analysis
van Houdt et al. [31]	General neural network review	Review of long short-term memory (LSTM) models
Wang et al. [32]	Prognostics for system health management	RUL estimation in system prognostics
Mao et al. [33]	Predictive maintenance of rolling bearings	RUL prediction of rolling bearings
Zhang et al. [3]	Operational research in degradation data analysis	Degradation data analysis for RUL estimation
Proposed	Prediction of bearing RUL under multi-damage and multi-working conditions	RUL prediction of mechanical bearings
**Reference**	**Data Preprocessing**	**Model Used**
Guo et al. [34]	Data normalization for time-series data	CNN-BiLSTM-Attention
Rincón-Maya et al. [35]	Feature extraction from current and voltage data	ICC-CNN-LSTM
Mejia et al. [36]	Filtering and normalization of time-series data	LSTM units with analysis filter bank
van Houdt et al. [31]	Not applicable (review article)	Review of LSTM architectures
Wang et al. [32]	Feature extraction and data scaling for system health data	Bidirectional LSTM (BiLSTM)
Mao et al. [33]	Deep feature extraction from sensor data	Deep feature representation + LSTM
Zhang et al. [3]	Wiener-process-based methods, degradation model estimation	Wiener-process-based methods
Proposed	Data normalization and data segmentation process data and extract features with CNN-Bi-LSTM	CNN-Bi-LSTM integration domain adaptation

**Table 2 sensors-24-06906-t002:** Datasets of IEEE 2012 PHM prognostic challenge.

Datasets	Operating Conditions		
	Conditions 1	Conditions 2	Conditions 3
Learning set	Bearing1_1	Bearing2_1	Bearing3_1
	Bearing1_2	Bearing2_2	Bearing3_2
Test set	Bearing1_3	Bearing2_3	Bearing3_3
	Bearing1_4	Bearing2_4	
	Bearing1_5	Bearing2_5	
	Bearing1_6	Bearing2_6	
	Bearing1_7	Bearing2_7	

**Table 3 sensors-24-06906-t003:** Number of sample points in datasets.

Dataset Name	Sample Size	Uncollected Sample
Bearing1_1	2803	0
Bearing1_2	871	0
Bearing1_3	2375	573 (5730 s)
Bearing1_4	1428	34 (339 s)
Bearing1_5	2463	161 (1610 s)
Bearing1_6	2448	146 (1460 s)
Bearing1_7	2259	757 (7570 s)
Bearing2_1	911	0
Bearing2_2	797	0
Bearing2_3	1955	753 (7530 s)
Bearing2_4	751	139 (1390 s)
Bearing2_5	2311	309 (309 s)
Bearing2_6	701	129 (1290 s)
Bearing2_7	230	58 (580 s)
Bearing3_1	515	0
Bearing3_2	1637	0
Bearing3_3	434	82 (820 s)

**Table 4 sensors-24-06906-t004:** Parameters of the CNN.

Experimental Parameter	Grid Search-Related Parameters			Final Numerical Value
Optimizer	Adam			Adam
Learning_rate	0.01	0.001	0.0001	0.01
Loss_function	CrossEntropyLoss			CrossEntropyLoss
Batch_size	8	16	32	32
Activation_function	ReLU	Leaky_ReLU	Tanh	ReLU
Number of iterations	10			200

**Table 5 sensors-24-06906-t005:** Experimental data and results.

Experimental Parameter	Numerical Value	Experimental Parameter	Numerical Value
Total_data	Condition 1 all data	Train_data	0.7 × total_data
Test_data_1	0.3 × total_data	Test_data_2	Condition 2 all data
Test_data_3	Condition 3 all data	Test_data_4	Condition 4 all data
Test_data_1_acc_rate	1.0	Test_data_2_acc_rate	1:1.0, 0:0.961
Test_data_3_acc_rate	1:1.0, 0:0.885	Test_data_4_acc_rate	1:1.0, 0:0.698

**Table 6 sensors-24-06906-t006:** CRWU dataset.

Breakdown	Condition 1			Condition 2			Condition 3			Condition 4		
	0.007″	0.014″	0.021″	0.007″	0.014″	0.021″	0.007″	0.014″	0.021″	0.007″	0.014″	0.021″
Inner	105.mat	169.mat	209.mat	106.mat	170.mat	210.mat	107.mat	171.mat	211.mat	108.mat	172.mat	212.mat
Ball	118.mat	185.mat	222.mat	119.mat	186.mat	223.mat	120.mat	187.mat	224.mat	121.mat	188.mat	225.mat
	130.mat	197.mat	234.mat	131.mat	198.mat	235.mat	132.mat	199.mat	236.mat	133.mat	200.mat	237.mat
Outer	144.mat		246.mat	145.mat		247.mat	146.mat		248.mat	147.mat		249.mat
	156.mat		258.mat	158.mat		259.mat	159.mat		260.mat	160.mat		261.mat
Normal	97.mat			98.mat			99.mat			100.mat		

**Table 7 sensors-24-06906-t007:** FOT of datasets.

Bearing	Sample Size	FOT ([9])	FOT (Amplitude)
1_1	2803	1490	1490
1_2	871	827	827
1_3	2375	1684	2200
1_4	1428	1083	1080
1_5	2463	680	2400
1_6	2448	649	2400
1_7	2259	1026	2200

**Table 8 sensors-24-06906-t008:** ACC of datasets.

Bearing	ACC_Rate ([9])	ACC_Rate (Amplitude)
1_1	0.9697	0.9697
1_2	0.9977	0.9977
1_3	0.7326	**0.8025**
1_4	0.8179	0.8172
1_5	0.3134	**0.9704**
1_6	0.4788	**0.7496**
1_7	0.4998	**0.9398**

**Table 9 sensors-24-06906-t009:** Basic parameters of the model.

Experimental Parameter	Random Search Numerical Value			Final Numerical Value
Optimizer	Adam			Adam
Learning_rate	0.01	0.001	0.0001	0.001
Loss_function	MSELoss			MSELoss
Num_layers	1	2	3	2
Hidden_size	64	128	256	64
Input_size	160			160
Batch_size	16	32	64	32
Number of iterations	10			200
Activation_function	ReLU	Leaky_ReLU	Tanh	ReLU

**Table 10 sensors-24-06906-t010:** Basic model evaluation metrics and evaluation indices for I-DCNN and MCNN.

Bearing	Proposed		I-DCNN		MCNN	
	MAE	RMSE	MAE	RMSE	MAE	RMSE
1_3	**0.0857**	**0.1120**	0.2190	0.2513	/	/
1_4	**0.1379**	**0.1927**	0.4865	0.5236	/	/
1_5	0.2128	0.2619	0.1949	0.2199	/	/
1_6	0.1848	0.2218	0.1734	0.2002	/	/
1_7	**0.1192**	**0.1587**	0.2145	0.2499	/	/
NRMSE	**0.1894**		0.2890		0.3805	

**Table 11 sensors-24-06906-t011:** Comparative analysis of the performance of various time-series models.

Bearing	Bi-LSTM		LSTM		RNN		Bi-RNN		GRU		Bi-GRU	
	NMAE	NRMSE	NMAE	NRMSE	NMAE	NRMSE	NMAE	NRMSE	NMAE	NRMSE	NMAE	NRMSE
OC1	**0.1201**	**0.1556**	0.2501	0.2888	0.2533	0.2943	0.2761	0.3308	0.2732	0.3250	0.1375	0.1804
OC2	0.2463	0.3019	0.2501	0.2888	0.2534	0.2945	0.2462	0.3310	0.2783	0.3295	0.2347	0.2889
OC3	**0.1664**	**0.2020**	0.2502	0.2890	0.2535	0.2945	0.2763	0.3311	0.3355	0.3860	0.1724	0.2138

**Table 12 sensors-24-06906-t012:** Jensen–Shannon divergence of baseline feature distribution.

Feature Region	JS Divergence (Bearing1_1)	JS Divergence (Bearing1_2)
Bearing1_1	/	0.1494
Bearing1_2	0.1494	/
Bearing1_3	0.6848	0.0837
Bearing1_4	0.4578	0.2097
Bearing2_1	0.4416	0.1500
Bearing2_2	0.6364	0.1249
Bearing3_1	0.8425	0.1065
Bearing3_2	0.5660	0.0663

**Table 13 sensors-24-06906-t013:** Basic parameters of the domain adaption experiment.

Experimental Parameter	Numerical Value
Optimizer	Adam
Learning rate	0.0001
Predict loss function	MSELoss
Domain loss function	CrossEntropyLoss
Num_layers	2
Hidden_size	64
Input_size	160
Batch_size	32
Number of iterations	200
Source-domain dataset	Bearing1_1
Target-domain dataset	Bearing2_1, Bearing2_2, Bearing3_1, Bearing3_2

**Table 14 sensors-24-06906-t014:** Domain adaption model and original model evaluation index.

Bearing	Domain_Model		Raw_Model	
	MAE	RMSE	MAE	RMSE
1_1	**0.0419**	**0.0578**	0.0511	0.0662
2_1	**0.2996**	**0.3572**	0.3007	0.3634
2_2	0.2027	0.2500	0.1964	0.2444
3_1	**0.1833**	**0.2274**	0.1880	0.2353
3_2	**0.1917**	**0.2274**	0.1955	0.2316

**Table 15 sensors-24-06906-t015:** Domain model evaluation metrics and evaluation indices for I-DCNN, MCNN, and CNN-Bi-LSTM.

Bearing	Proposed		I-DCNN		MCNN		CNN-Bi-LSTM	
	MAE	RMSE	MAE	RMSE	MAE	RMSE	MAE	RMSE
1_3	**0.0818**	**0.1061**	0.2190	0.2513	/	/	0.0857	0.1120
1_4	**0.1373**	**0.1916**	0.4865	0.5236	/	/	0.1379	0.1927
1_5	0.2198	0.2713	0.1949	0.2199	/	/	0.2128	0.2619
1_6	0.1765	0.2115	0.1734	0.2002	/	/	0.1848	0.2218
1_7	**0.1180**	**0.1549**	0.2145	0.2499	/	/	0.1192	0.1587
NRMSE	**0.1870**		0.2890		0.3805		0.1894	

## Data Availability

The CRWU dataset and the PHM2012 Challenger dataset are used in this article (two versions are available, compressed and uncompressed). Related datasets are uploaded onto Google cloud disk; link is as follows: https://drive.google.com/drive/folders/1bXvElCVCr0smjgKW4a0feG5L8W9U08sq?usp=drivel (accessed on 18 September 2024). We have also uploaded the experimental code to GitHub. The link is https://github.com/FeiFanLi-arch/phm2012_domain_cnn_bi_lstm.git (accessed on 18 September 2024). Detailed information is listed in the README file.

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
