# Peer review of "Prediction of the Remaining Useful Life of Bearings Through CNN-Bi-LSTM-Based Domain Adaptation Model"

_sensors, 2024, doi:10.3390/s24216906_

Round 1

Reviewer 1 Report

Comments and Suggestions for Authors

This manuscript proposes a CNN-Bi-LSTM model framework, in which CNN extracts signal features in discrete time and Bi LSTM captures the temporal features of these discrete time features, and then integrates a domain adaptation module to improve the model's generalization performance. This work has a certain significance. However, there are still some issues.

1.What is the logical relationship between the fourth and fifth paragraphs of Introduction?

2. The methods proposed by the author are quite common, whether they are convolutional neural networks (CNN) or bidirectional long short-term memory networks (Bi LSTM). Is this just a new application of these methods in another field? It is necessary to clarify the advantages of the combination of CNN and Bi LSTM in predicting bearing RUL compared to existing related technologies

3. How does the domain adaptation module work? Why can this module minimize feature differences. Regarding this issue, it seems that 2.1.7 in the article did not provide a clear explanation. Explain how domain adaptation techniques are integrated into the model and how their work affects model performance

4. We suggest adding comparative experiments with the latest methods to demonstrate the effectiveness of the proposed method and increase the persuasiveness of the article.

5. Some figures qualities are hope to be improve in this manuscript, such as Fig 7,8,9, and Fig6.

Author Response

Please see the attachment.A template can be found here.

Reviewer 2 Report

Comments and Suggestions for Authors

Thank you for the opportunity to review this manuscript. This paper addresses a CNN-Bi-LSTM model, extracts signal features, captures temporal patterns, and predicts RUL using spatiotemporal data, which is both relevant and timely in the field of condition-based maintenance and prognostics and health management. The research proposes a deep-learning-based model for predicting bearing life.  Suggestions and observations that may help strengthen the paper further, can be found in the attached document.

Reviewer 3 Report

Comments and Suggestions for Authors

This work is focuses on the proposal of a neural network-based feature engineering and the downstream prediction of the RUL, certainly, the proposal is based on CNN and Bi-LSTM. The achieved results are promising but some issues have to be addressed.

1. The literature review is very restrictive, most of the discussed works are focused only in deep learning approaches; however there are important machine learning strategies that can be also discussed. Additional, other deep learning algorithms such as auto encoder can be also discussed in the introduction section. Please try to complete the literature review and consider the following references:: https://doi.org/10.1177/14759217231221214 ; https://doi.org/10.3390/s21175832 ; https://doi.org/10.1016/j.measurement.2023.112774

2. Figure 1 aims to depict the proposed methodology, however there is no relationship with titles of all subsections in section 2; in order to be more clear, it is suggested to name subsection similarly than names included in figure 1 diagram.

3 The results are interesting but are not compared.with. Other previous methods, try to include a comparison.

4. Regarding the computational burden, can the authors include detailed information?

5 what are the main limitations of this proposal, include a brief description in the conclusion section.

Reviewer 4 Report

Comments and Suggestions for Authors

This paper put forth a neural network-based methodology for forecasting the remaining useful life (RUL) of a system. The proposed approach employed a convolutional neural network (CNN) to extract pertinent features from data points and a bidirectional long short-term memory (Bi-LSTM) network to capture temporal relationships among the data. This method effectively supplanted traditional feature engineering techniques that relied on prior knowledge. However, some sections of the article require revision.

1.      “This approach assisted the model in extracting and representing various feature types.” Please add a reference here.

2.      “Existing bearing prediction methods can be broadly categorized into two types: traditional machine learning methods that incorporate prior knowledge and deep learning methods primarily driven by data.” Please add a reference here.

3.      “1D convolutional layers possess strong nonlinear extraction capabilities and are widely used in processing 1d sequence data, such as text classification, speech recognition, signal processing, and time-series analysis.” Please add a reference here.

4.      “This approach helps the model to adapt the features from the source domain to match those of the target domain, while minimizing feature discrepancy.” Please add a reference here.

5.      “Most of the traditional RUL prediction methods preprocess the data through feature engineering, extract the time-frequency domain features of the data according to relevant prior knowledge, and then predict the bearing RUL after model training.” Please add a reference here.

6.      “This method can reduce the cost of feature engineering without selecting time-frequency domain features according to relevant prior knowledge, so that the model can be better applied to different data sets and then applied in industry.” Please add a reference here.

7.      “At the same time, bearing data belongs to time sequence data, and time sequence information is hidden between points, and the Bi-LSTM model has certain advantages in extracting time sequence information.” Please add a reference here.

8.      “The formula of MAE is shown in (Eq.2), and the formula of RMSE is shown in (Eq.3).” Please add a reference here.

9.      “It is particularly useful for applications where each error is equally important, as it does not square the differences,” Please add a reference here.

10.  “Therefore, RMSE is often preferred when large errors are particularly undesirable, and reducing them is a priority.” Please add a reference here.

11.  “Both metrics provide complementary perspectives: MAE gives a straightforward view of overall error, while RMSE emphasizes the significance of larger mistakes in the prediction.” Please add a reference here.

12.  This paper first uses a convolutional neural network (CNN) model for feature engineering, and then uses a bidirectional long short-term memory network (Bi-LSTM) model to capture the time-series degradation characteristics of the engineered features and predict the RUL by regression. The logic for using the two models should be clarified more.

13.  It is preferable to include a discussion section that explains the significant advances in approach and understanding over previous literature and/or convincingly demonstrates the potential for new applications.

14.  The conclusion should be rewritten and some quantitative results should be added.

15.  It would be beneficial to include some background information on the life cycle of mechanical bearings.

Round 2

Reviewer 1 Report

Comments and Suggestions for Authors

All the issues I am concerned about have been resolved well, there are no other issues

Author Response

Comments 1: [All the issues I am concerned about have been resolved well, there are no other issues.]

Response 1: [Thank you for your suggestions and comments, as well as your recognition.]

Reviewer 2 Report

Comments and Suggestions for Authors

I appreciate the second version; there was a significant improvement between the first and second versions of the article. However, many of the additions should be incorporated directly into the text.

Reviewer 4 Report

Comments and Suggestions for Authors

I think this manuscript can be accepted in this form.

Author Response

Comments 1: [I think this manuscript can be accepted in this form.]

Response 1: [Thank you for your suggestions and comments, as well as your recognition.]